# Behaviour of *Escherichia coli* O157:H7 and *Listeria monocytogenes* in Normal and DFD Beef of an Autochthonous Portuguese Breed

**DOI:** 10.3390/foods12071420

**Published:** 2023-03-27

**Authors:** Cristina Saraiva, Sónia Saraiva, Luis Patarata, Maria da Conceição Fontes, Conceição Martins

**Affiliations:** 1Animal and Veterinary Science Center (CECAV), University of Trás-os-Montes e Alto Douro (UTAD), 5000-801 Vila Real, Portugal; 2Department of Veterinary Sciences, School of Agricultural and Veterinary Sciences, UTAD, 5000-801 Vila Real, Portugal; 3Associate Laboratory for Animal and Veterinary Science (AL4AnimalS), 1300-477 Lisboa, Portugal

**Keywords:** autochthonous breed, beef, DFD meat, *Listeria monocytogenes*, *Escherichia coli* O157:H7, ultimate pH

## Abstract

This study was carried out to identify the behaviour of *Escherichia coli* O157:H7 and of *Listeria monocytogenes* inoculated in Maronesa breed beef with different ultimate pH (pHu) (Normal and DFD), and stored at two different temperatures (4 and 9 °C), during 28 days *post mortem* (pm). The main objective was to illustrate the problematic feature of dealing with beef showing high pHu and stored at mild abusive temperatures (9 °C). Beef steaks (ms. *longissimus dorsi*) were inoculated with low levels (2–3 log CFU/g) of those both pathogens and packed in air, vacuum and three gaseous mixtures with decreasing O_2_ and increasing CO_2_ concentrations (MAP70/20, MAP50/40 and MAP30/60). At 4 °C, the growth of *E. coli* O157:H7 presented the same pattern on Normal and DFD meat. On the contrary, the growth of *L. monocytogenes* was higher in DFD meat, revealing the effect of the pHu and its psychotropic character. At abusive temperatures, both pathogens grew, achieving high levels in DFD meat. In these cases, the MAP with the highest CO_2_ concentration (60%) was revealed to be more effective against the development of *E. coli* O157:H7, therefore, not exceeding levels of 5 log CFU/g at the end of storage, while in *L. monocytogenes,* it reaches 8 log CFU/g under the same conditions.

## 1. Introduction

Beef from the Maronesa cattle has high commercial value due to its autochthonous breed, environmentally sustainable production system, and it has Protected Designation of Origin (PDO) [1,2]. However, in order to exploit the full potential of meat products, it is important to ensure that products reach the consumer in the best condition [3]. The quality of the meat can be affected by intrinsic and extrinsic parameters, such as age, sex, breed [4], feeding [5], enzyme activity [6], meat-modified atmosphere packaging (MAP) [7], or meat contamination [8]. The hygienic quality, which is a result of beef processing and handling, is particularly important because it has direct implications for consumer confidence [9]. Contamination of beef occurs throughout slaughtering, deboning, cutting, packaging, and storage, during which time temperature and hygiene are important parameters affecting the shelf-life of fresh meat for industrial use or retail [10,11].

Microbial pathogens are usually associated with the most serious meat safety issues in product recalls and foodborne diseases [12,13,14]. Nowadays, besides the considerable knowledge about the microbiota responsible for foodborne diseases, a high number of outbreaks and incidents keep occurring, many of them associated with meat and meat products which in 2021 represented 11.9% of positive units reported by the MS of EU [15]. Beef, in particular, has been associated with foodborne diseases, causing tragic outcomes involving death in some cases. An aiding fact of foodborne infections due to beef consumption may be related to its preparation methods, in which these meats are many times only subjected to mild heat treatments, commonly referred to as undercooked or rare [16]. Enterohemorrhagic *E. coli* O157:H7 is a highly pathogenic subset of Shiga toxin-producing *E. coli* [17]. *E. coli* belongs to the enteric microflora of many healthy animals, with cattle being the main reservoirs of *E. coli* O157:H7 [18,19]. *E. coli* O157:H7 has emerged as an important foodborne pathogen that causes diarrhoea, hemorrhagic colitis, and hemolytic-uremic syndrome in humans [17] and, in severe cases, can cause death [20]. On the other hand, *L. monocytogenes* has a ubiquitous distribution and a great ability to grow in a wide range of conditions, such as refrigeration [21,22] and has been regarded as the pathogen of concern in ready-to-eat (RTE) meat [23]. *L. monocytogenes* is also one of the most serious agents of foodborne diseases under EU surveillance, and infection in humans is still high, some of them with death [15]. Unsafe practices, including long storage duration and abusive temperatures, have a potential impact on human listeriosis risk [24]. The temperature of domestic refrigerators was shown to be very variable through a review of survey studies from 1991 to 2016. It was observed that mean temperatures ranged from <5 to 8.1 °C, the minimum temperatures from −7.9 to 3.8 °C and the maximum from 11.4 to 20.7 °C [24].

The gaseous composition of the packaging and pHu values are parameters that can affect the growth of microorganisms in meat [25,26]. In red meat products, one of the most common methods of packaging is the use of MAP, usually through mixtures of CO_2_, O_2_ and N_2_, each one with a specific function. The MAP generally inhibits microbial spoilage of fresh meat, minimising the loss of products and maintaining a higher quality in perishable food during its normal shelf-life or extending it [27]. Nevertheless, MAP should be associated with strict temperature control to achieve maximum microbial inhibition [11,28]. The pHu of meat higher than the normal values (pHu: 5.5–5.8), as in dark, firm and dry (DFD) (pHu: >6.1) or moderate DFD (pHu: 5.9–6.1) meat, can be responsible for the meat spoilage [29,30]. The production system of Maronesa beef is known for the occurrence of DFD conditions [31]. Consequently to the DFD condition, a reducing of the product’s shelf-life can occur and it is also less acceptable to the consumer [31,32]. According to Silva et al. [31], during two years of measurements of pHu, it was noticed that in *longissimus dorsi,* about 25% of the cases had a final pH equal or superior to 6.2 and were classified as DFD. Thus, the results revealed that the occurrence of high-pH meat is dependent on the muscle. Other muscles, such as *gracillis* (8%) and *psoas major* (≤2%) muscles, presented with a lower percentage of DFD condition. DFD meat is related to animal stress and transport with multifactorial origins. It has been reported in many countries, with variable occurrence rates of 1.3% in Canada, 3.2% in the USA, 4.5% in Brazil and 13.45% in Mexico [29,33,34,35]. The incidence of DFD cases was also dependent on the sex of the animal, with a higher occurrence of DFD cases observed in males [31].

Considering the lack of studies related to the quality of autochthonous breeds meat associated with the occurrence of DFD beef in the Maronesa breed, the aim of this study was to evaluate the influence of pHu (Normal and High) and meat packaging (air, vacuum and three MAP with gas) on the behaviour of *E. coli* O157:H7 and *L. monocytogenes* inoculated on beef of Maronesa breed and stored at 4 ± 0.5 °C and 9 ± 0.5 °C, during 28 days of storage.

## 2. Materials and Methods

### 2.1. Experimental Design

#### 2.1.1. Sampling

Beef *longissimus dorsi* muscle was obtained from eight Portuguese autochthonous Maronesa, 9 to 11-month-old bulls whose carcasses weighed from 90 to 150 kg. *Longissimus dorsi* was excised from the carcasses between the sixth thoracic and the second lumbar vertebra at 24 h post-mortem (pm). Based on pHu measured at 24 h pm directly in the muscle using a combined glass electrode with a pH meter Crison 2002, muscles were divided into two pH groups: Normal (pH ≤ 5.8, n = 4) and DFD (pH ≥ 6.2, n = 4). After that, muscles were cut into pieces of approximately 200 g, packed in a vacuum and stored at −80 °C until the beginning of the experiment. On day 1 of the experiment, cuts of muscles were kept at 2 °C for 2 h. After this time, approximately 1 cm of the external surface of the meat was aseptically removed, and cuts were sliced. At the end of this process, pieces of meat (0.5 cm thick, surface 2 × 2.5 cm, ≈5 g) were obtained. Immediately after this preparation, meat samples were analysed (24 h pm) for meat characterisation and discarded *L. monocytoge*nes and *E. coli* O157:H7 contamination prior to inoculation. Experiments were performed using four animals in each experimental unit per each pH group, and a control sample was prepared in all conditions of the experimental design.

#### 2.1.2. Microorganisms and Growth Conditions

Pieces of meat were inoculated with 20 µL of a suspension of *E. coli* O157:H7 (NCTC 12900) and *L. monocytogenes* (ATCC 7973) for an inoculation level of 2–3 log (CFU/g) per strain and package.

Inoculates used in this study were prepared at growth suspension of *L. monocytogenes* (30 °C, 24 h) and *E. coli* O157:H7 (37 °C, 24 h) in brain heart infusion broth (Oxoid CM225). Cells were centrifugated (5000× *g*, 15 min, 4 °C) and washed three times in 0.85% sterile physiologic saline and compared to a 0.5 McFarland turbidity standard. Serial (10-fold) dilutions were performed to yield approximately 1 × 10^3^ cells/cm^2^. To verify the number of viable *L. monocytogenes* and *E. coli* O157:H7 in the suspension, dilutions were spread on Compass *L. monocytogenes* Agar (Biokar BM06508) and CT-SMAC (Biokar BK147 + BS037), respectively.

#### 2.1.3. Packaging

Inoculated and control samples were packed in five different types of packaging, namely: air; vacuum; 70%O_2_/20%CO_2_/10%N_2_ (MAP70/20); 50%O_2_/40%CO_2_/10%N_2_ (MAP50/40); and 30%O_2_/60%CO_2_/10%N_2_ (MAP30/60).

In air packaging, meat pieces were tray-packaged in air overwrapped with polyethylene film and in a vacuum: they were individually vacuum packaged in COMBITHERM bags (WIPAK Walsrode, HAFRI) which have an oxygen transmission rate (OTR) of 63 cm^3^ m^−2^d^−1^atm^−1^ at 23 °C, 0% RH and water vapour transmission (WVT) of 1 g m^−2^d^−1^ at 23 °C, 85% RH. For MAP, pieces of meat were individually placed in COMBITHERM XX bags (WIPAK Walsrode, HAFRI) 0.115 mm thick and OTR of 1 cm^3^ m^−2^d^−1^atm^−1^ at 23 °C, 0% RH and WVT of 1 g m^−2^d^−1^ at 23 °C, 85% RH. The atmosphere in the MAP was first removed and then flushed with the appropriate gas mixture (Praxair, Portugal) using a SAMMIC V-420 SGA. The final ratio between gas and meat was approximately 3:1.

Samples were stored at 4 ± 0.5 °C and 9 ± 0.5 °C and examined for microbiological counts on days 1 (2 h after packaging), 3, 7, 10, and 14 pm. For air packaging, microbiological counts were not carried out at 14 days pm. On the contrary, on vacuum packaging, microbiological counts were also carried out at 21 and 28 days pm.

### 2.2. Microbial Analysis

Meat pieces were aseptically collected at each interval. Samples were homogenized with sterile buffered peptone water for 90 s in a Stomacher (IUL, Barcelona, Spain). Serial decimal dilutions were prepared in the same solution and were plated on CT-SMAC (Biokar BK147 + BS037) for *E. coli* O157:H7 counts (37 °C for 24 h) and Compass L. mono agar (Biokar BM06508) in case of *L. monocytogenes* (37 °C for 24–48 h). After incubation, typical colonies were counted, and results were expressed in log CFU/g.

### 2.3. Data Analysis

One-way analysis of variance (ANOVA) was conducted to test the effect of pHu (Normal and DFD) and of the temperature of storage (4 ± 0.5 °C and 9 ± 0.5 °C), for each day of microbiological counts (1, 3, 7, 10, 14, 21, 28 days pm) using the SPSS 22.0 software (SPSS, IBM, Armonk, NY, USA) at 5% level of probability.

## 3. Results

### 3.1. Behaviour of Escherichia coli O157:H7

The results of counts of *E. coli* O157:H7 in inoculated beef samples, according to pHu (Normal and DFD) and the type of packaging for each storage temperature, are presented in Table 1. Two hours after inoculation, the Normal and DFD samples had a similar count for *E. coli* O157:H7. These counts were slightly greater than the programmed level of inoculation, though this was a very slight excess and never attained a logarithmic unit. The final pH of the meat was not decisive on the growth of *E. coli* O157:H7 in beef stored at 4 ± 0.5 °C. The highest *E. coli* O157:H7 counts for DFD meat and temperature of 9 °C were observed in vacuum packaging for all storage times (4.39 ± 0.65 log CFU/g at 3 days pm; 6.57 ± 0.26 log CFU/g at 7 days pm; 7.77 ± 0.32 log CFU/g at 10 days pm and 8.35 ± 0.21 log CFU/g at 21 days pm). However, no significant differences were observed between Normal and DFD pHu at 3 days pm. At 10 days pm, the largest difference between the DFD and Normal meat was observed in the vacuum atmosphere (4.05 log CFU/g; *p* < 0.001) followed by the air atmosphere (3.61 log CFU/g; *p* < 0.01). In the MAP70/20, differences between the DFD and Normal meat reached 3.04 log CFU/g (*p* < 0.001). The MAP packaging with more concentration of CO_2_, namely MAP30/60, showed the lowest counts with no significant differences between pH groups at 10 and 14 days pm.

The counts of *E. coli* O157:H7 on Normal and DFD beef were similar at a temperature of 4 °C during the storage period, maintaining the initial levels. However, at 9 °C, growth levels were achieved for DFD meat of ~8 log CFU/g in vacuum from day 10 pm, reaching the highest value of 8.76 ± 0.64 log CFU/g at day 28 pm. In the MAP with the high CO_2_ concentration (MAP30/60), counts were not higher than 5 log CFU/g, which appears to reveal the inhibitory effect of CO_2_ in the development of *E. coli* O157:H7 (Figure 1). The pathogen *E. coli* O157:H7 packed in MAP70/40 and MAP30/60 and stored at mild abusive temperature (9 °C) followed the same pattern and showed similar counts over time.

### 3.2. Behaviour of Listeria Monocytogenes

The results of counts of *L. monocytogenes* in inoculated beef samples, according to pHu (Normal and DFD) and the type of packaging for each storage temperature, are presented in Table 2. At 1 day pm, in the five packagings, DFD and Normal samples had a similar count for *L. monocytogenes* with a score closer to that desired in all packaging. Two more days after storage, the consequences of temperature started to show that *L. monocytogenes* increased considerably in DFD compared to Normal meat, and these differences were more pronounced, particularly in MAP70/20 compared to the other two MAP. *L. monocytogenes* achieved at 3 days pm and 7 days pm average counts of 3.41 ± 0.92 log CFU/g and 6.49 ± 0.83 log CFU/g in DFD meat and 2.61 ± 0.65 log CFU/g and 1.74 ± 0.76 log CFU/g in Normal meat, which generally maintained the same pattern during the remaining storage time.

In Normal pH, the effect of abusive storage temperature was not noticed in the development of *L. monocytogenes*, contrarily to DFD meat which presented very significant differences in most situations in which the meat was in abusive temperatures.

The MAP30/60, with the higher content of CO_2,_ seems to inhibit this microorganism a little more, showing inferior counts for all storage times when compared with MAP70/20 and MAP50/40.

At a temperature of storage of 4 °C, the growth of *L. monocytogenes* was not observed for Normal meat. On DFD meat, the counts achieved levels of 6 log CFU/g, 7 log CFU/g and 8 log CFU/g in vacuum at day 14 pm, 21 pm and 28 pm, respectively. The lowest counts were obtained on meat packaged in the MAP30/60 for a long time.

At abusive temperature (9 °C) and Normal pHu, *L. monocytogenes* presented similar counts during the storage period. On the contrary, for DFD meat, *L. monocytogenes* developed very quickly after day 3 pm, achieving values higher than 8 log CFU/g in air and vacuum at day 10 pm. These levels were attained in MAPs later at day 14 pm.

## 4. Discussion

The pHu and storage temperature significantly influenced the growth of both *E. coli* O157:H7 and *L. monocytogenes.*

At 4 °C of storage, the counts of *L. monocytogenes* for all packages in DFD beef were below 5 log CFU/g (10 days pm), with the higher counts observed in the air (4.47 log CFU/g; *p* < 0.05) and vacuum (4.55 log CFU/g; *p* < 0.05) packages (Table 2). On the contrary, at 4 °C the counts of *E. coli* H157:H7 were about 3 log CFU/g during all times of storage (3, 7, 10, 14, 21 and 28 days pm) in all types of packaging, showing no significant differences (Table 1). Thus, the low storage temperature (4 °C) associated with the DFD condition was not enough to produce an extensive inhibition of *L. monocytogenes* as observed in *E. coli* O157:H7. These results can be justified by the fact that *L. monocytogenes* is a psychrotrophic bacterium, capable of surviving and multiplying at low temperatures, both under aerobic and anaerobic conditions and adhering to various surfaces [36]. Moreover, *L. monocytogenes* has the ability to grow at a pH of 6 or higher, as observed in DFD meat [37] and has a high tolerance to low pH and high salt concentration [38]. According to Nissen et al. [39], at 4 °C, *L. monocytogenes* and *Y*. *enterocolitica* are considered the most serious pathogens in meat. Low temperatures induce enzymes such as RNA helicase, which improves the activity of *L. monocytogenes*, as well as replication at low temperatures. Moreover, the capacity to produce biofilms enhances *L. monocytogenes’* ability to survive harsh environments and also use flagella at lower temperatures which enables the ability to propel itself [40]. Elevated CO_2_ and reduced O_2_ levels are commonly used to extend the shelf-life of food products through the inhibition of microbial growth and oxidative changes [41]. The use of O_2_-free atmospheres in packages has been suggested for different meat products [42]. In the present study, MAPs inhibited, compared to the air and vacuum atmospheres, the development of *L. monocytogenes* mainly in the MAP with the highest concentration of CO_2_. The lower OTR (1.0 cm^3^m^−2^day^−1^) of the packaging film used in this experiment for MAP, in comparison to films of greater O_2_ permeability (OTR = 4.5 cm^3^m^2^day^−1^) used in the study conducted by Tsigarida et al. [43] can justify these results. Saraiva et al. [41], using the same packaging film as the present study, found that *L. monocytogenes* in beef may be reduced by ~1.0 log in vacuum packaging and by ~1.5 log on average in the MAPs. Generally, under an anaerobic-modified atmosphere, the LAB compete with the support microflora and have shown to be effective in inhibiting the growth of pathogenic bacteria such as *L. monocytogenes* in meat products [44]. The combination of selected LAB strains with antimicrobial compounds, for instance, acid and sodium lactate or the use of active packaging, could be the next step strategy for eliminating the risk of *L. monocytogenes* in meat and dairy-ripened products [45]. Many food-spoiling LABs are facultative aerobic and quite resistant to CO_2_. This contributes to the fact that LABs can be found as the main spoilers on high-oxygen packaged meat [41]. Therefore, *L. monocytogenes* can represent a risk when stored at temperatures higher than 0 °C, and the efficacy of thermal treatments is limited by the ability to survive and actively replicate at temperatures between −0.4 and 45 °C [46]. Air and vacuum atmospheres had similar counts for this pathogen (Figure 2), which could be related to the O_2_ permeability rate of the packaging film used for the vacuum atmosphere (OTR = 63 cm^3^m^−2^day^−1^), not so efficient in inhibiting the *L. monocytogenes* [41].

A higher proportion of CO_2_ in MAP50/40 and even more in MAP30/60 have led to lower growth of *L. monocytogenes* compared to other types of packaging. This result is consistent with other research whose results confirmed that the growth of *L. monocytogenes* was inhibited with increasing concentrations of CO_2_ [39,47]. According to Saraiva et al., [41] *L. monocytogenes* requires CO_2_ levels of 40% *v/v* or higher for effective inhibition. In view of maintaining the cytoplasmic pH within a range that is consistent with growth and survival, the decarboxylation reaction needs to occur to help maintain the cytoplasmic pH, though a high concentration of CO_2_ can inhibit the decarboxylation reaction by which CO_2_ is released through feedback mechanisms [41,48].

At a temperature of storage of 4 °C, the counts of *E. coli* O157:H7 on Normal and DFD, as well as for abusive temperature on Normal beef, were similar during the storage period, maintaining the initial levels (Figure 1). Regarding DFD meat and temperature of storage of 9 °C, the *E. coli* O157:H7 achieved the highest growth levels in a vacuum atmosphere (~8 log CFU/g), showing statist differences from Normal meat (*p <* 0.001) and from a temperature of storage of 4 °C (*p <* 0.001) at 7, 10, 14, 21 and 28 days pm.

The pathogen *E. coli* O157:H7 in air packaging multiplied rapidly; however, MAP30/60 showed lower growth levels during the storage period. In this study, this was the atmosphere with high CO_2_ content (60%), and *E. coli* O157:H7 counts did not attain 5 log CFU/g at 14 days pm. These results can be due to the inhibitory effect of CO_2_. In previous experiments, *E. coli* O157:H7 in atmospheres with high concentrations of CO_2_ (ratio of 30%CO_2_:70%O_2_ or 0.4%CO:60%CO_2_:39.6%N_2_) have been reported as being inhibitory to the multiplication *of E. coli* O157:H7 at an abusive storage temperature of 10 °C [38]. As mentioned above, for *L. monocytogenes*, *E. coli* uses the same mechanism of decarboxylation systems to protect the cell from a precipitous drop in pH. These systems depend on the activity of cytoplasmic pyridoxal-5′-phosphate (PLP)-containing amino acid decarboxylases, which consume one proton and release one CO_2_ for every molecule of substrate amino acid, thus helping maintain the cytoplasmic pH [49]. At refrigeration temperatures from 0 to 2 °C, high concentrations of CO_2_ (ratio of 20%CO_2_:80%O_2_ or 0.4%CO:30%CO_2_:69.6%N_2_) can lead to a reduction of one logarithmic unit in the levels from *E. coli* O157:H7 [50]. In accordance with the present study, Nissen et al. [39] reported that *E. coli* O157:H7 could be inhibited at 10 °C at high CO_2_ concentration and pHu inferior to 6. Moreover, even in MAP with a high CO_2_ and low CO mixture, meat acquires a stable colour, and the shelf-life can be extended.

The behaviour of *L. monocytogenes* in abusive temperatures differed between DFD and Normal pHu meats for all types of packaging after 7 days pm. For instance, at 10 day pm, Normal and DFD meat showed, respectively, counts of 2.07 log CFU/g and 8.25 log CFU/g in the air (*p <* 0.001); 2.54 log CFU/g and 8.24 log CFU/g (*p <* 0.001) on Vacuum; 1.69 log CFU/g and 7.91 log CFU/g in MAP70/20 (*p <* 0.001); 1.37 log CFU/g and 7.79 log CFU/g in MAP50/40 (*p <* 0.001); and 1.42 log CFU/g and 7.01 log CFU/g in MAP30/60 (*p <* 0.001). However, regarding the stored temperatures, for instance, at 14 day pm, count obtained at 4 °C and 9 °C of storage for DFD meat were respectively 5.71 log CFU/g and 8.33 log CFU/g (*p* < 0.001) in Vacuum; 4.82 log CFU/g and 8.55 log CFU/g in MAP70/20 (*p* < 0.001); and 4.07 log CFU/g and 8.53 log CFU/g in MAP50/40 (*p <* 0.001). Thus, even with meat stored at temperatures within recommended limits, the development of *L. monocytogenes* occurred markedly in meat with a high pHu, though when comparing both temperatures, the effect was not as intense as was the effect of pHu. However, unlike *E. coli* O157:H7, the effect of pH was not felt when storage was carried out at the appropriate temperature. Also, in this pathogen, at 9 ± 0.5 °C of storage, significantly greater growth was noted in the DFD meat samples.

## 5. Conclusions

The present study highlights the importance of maintaining the cold chain under strict surveillance conditions is confirmed to avoid abuses, which can have even more serious consequences in the case of DFD meat. As referred to above, the occurrence of DFD condition was more frequently observed in the *L. dorsi* muscle of the Maronesa breed, mainly in males. The results revealed that this condition associated with abusive storage temperatures allowed the growth of *E. coli* O157:H7 at higher levels, which is especially more evident in air and vacuum packages. For *L. monocytogenes,* the low storage temperature (4 °C) associated with the DFD condition was not enough to inhibit the growth of *L. monocytogenes,* as observed in *E. coli* O157:H7. This is due to the characteristic psychotropic of *L. monocytogenes*.

The use of special packing conditions, such as vacuum or MAP, in order to increase the shelf-life of meat or for technological and sensorial reasons can have consequences in terms of the development of some of the pathogens present in meat. Overall, the MAP were the most effective in controlling the development of *E. coli* O157:H7 and *L. monocytogenes*. In MAP, the effect of the CO_2_ level was observed, being even more noticeable in pH that better supported microbial growth.

## Figures and Tables

**Figure 1 foods-12-01420-f001:**
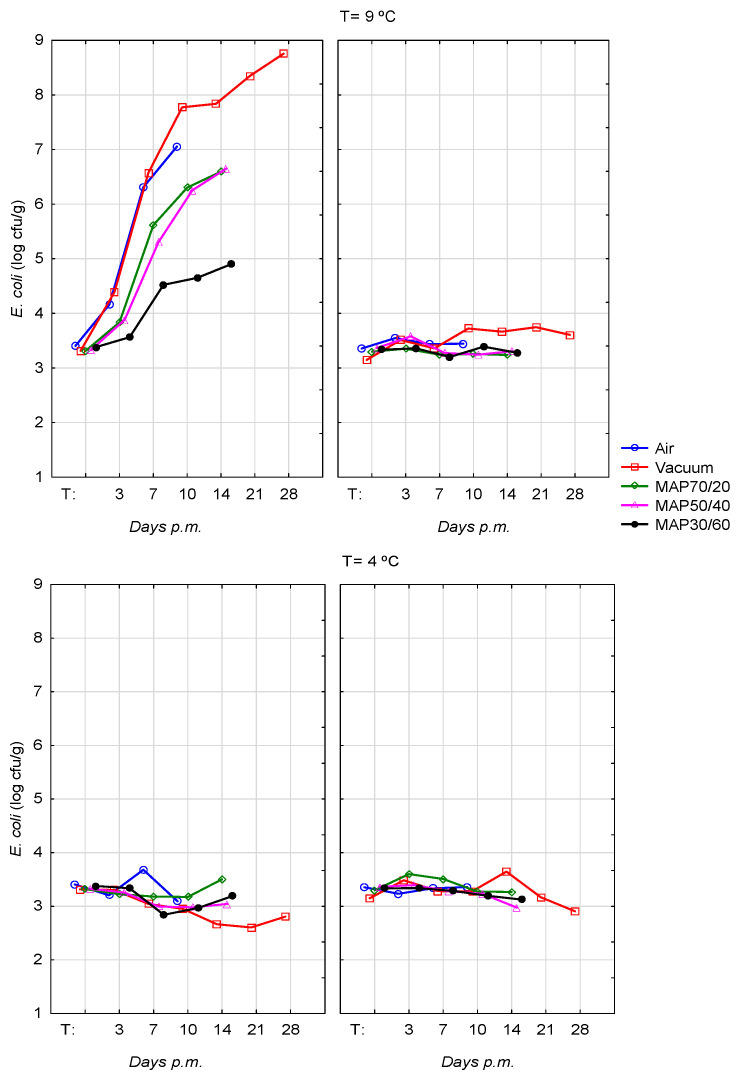
Evolution of *E. coli* O157:H7 levels (log UFC/g) on DFD (left) and Normal (right) beef stored at 4 °C and at 9 °C, packed in air, vacuum and MAPs -O_2_/CO_2_/N_2_- 70/20/10 (MAP70/20), 50/40/10 (MAP50/40), 30/60/10 (MAP30/60), during the time of storage.

**Figure 2 foods-12-01420-f002:**
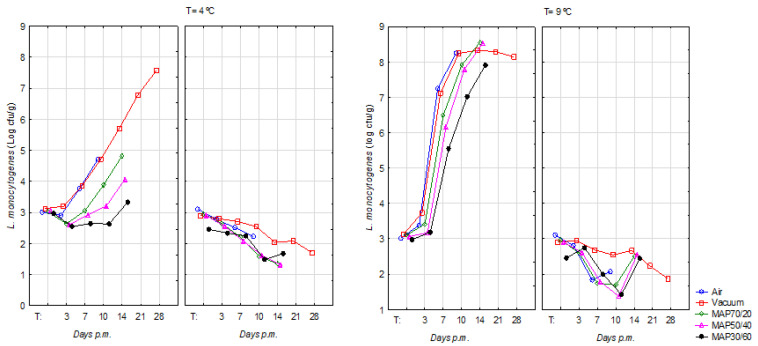
Evolution of *L. monocytogenes* levels (log UFC/g) on DFD (left) and Normal (right) beef stored at 4 °C and at 9 °C, packed in air, vacuum and MAPs -O_2_/CO_2_/N_2_- 70/20/10 (MAP70/20), 50/40/10 (MAP50/40), 30/60/10 (MAP30/60), during time of storage.

**Table 1 foods-12-01420-t001:** Counts of *E. coli* O157:H7 (log CFU/g) in inoculated beef samples according to pH groups (Normal and DFD) in five different packagings and stored at two temperatures (4 ± 0.5 °C and 9 ± 0.5 °C) at day 1, 3, 7, 10, 14, 21 and 28 pm.

Days pm	Temp. (°C)	Air			Vacuum			MAP70/20			MAP50/40			MAP30/60		
		N **	DFD	Sig.	N	DFD	Sig.	N	DFD	Sig.	N	DFD	Sig.	N	DFD	Sig.
1	A.I. *	3.35 ± 0.26	3.41 ± 0.32	ns	3.15 ± 0.52	3.31 ± 0.37	ns	3.30 ± 0.20	3.32 ± 0.25	ns	3.36 ± 0.37	3.33 ± 0.37	ns	3.33 ± 0.29	3.37 ± 0.33	ns
3	4 ± 0.5	3.23 ± 0.78	3.20 ± 0.42	ns	3.48 ± 0.27	3.30 ± 0.33	ns	3.60 ± 0.18	3.23 ± 0.27	ns	3.39 ± 0.46	3.25 ± 0.27	ns	3.34 ± 0.26	3.33 ± 0.26	ns
	9 ± 0.5	3.55 ± 0.45	4.16 ± 1.15	ns	3.51 ± 0.48	4.39 ± 0.65	ns	3.35 ± 0.43	3.84 ± 0.29	ns	3.58 ± 0.27	3.88 ± 0.56	ns	3.36 ± 0.27	3.56 ± 0.30	ns
	Sig.	ns	ns		ns	*		ns	*		ns	ns		ns	ns	
7	4 ± 0.5	3.33 ± 0.21	3.68 ± 0.76	ns	3.28 ± 0.29	3.04 ± 0.30	ns	3.51 ± 0.26	3.18 ± 0.21	ns	3.28 ± 0.19	2.99 ± 0.37	ns	3.28 ± 0.16	2.84 ± 0.23	*
	9 ± 0.5	3.43 ± 0.73	6.31 ± 1.26	**	3.36 ± 0.42	6.57 ± 0.26	***	3.24 ± 0.30	5.61 ± 0.12	***	3.27 ± 0.35	5.30 ± 0.49	***	3.20 ± 0.35	4.52 ± 0.78	*
	Sig.	ns	*		ns	***		ns	***		ns	***		ns	**	
10	4 ± 0.5	3.36 ± 0.25	3.09 ± 0.60	ns	3.28 ± 0.36	2.95 ± 0.43	ns	3.27 ± 0.24	3.17 ± 0.42	ns	3.23 ± 0.19	2.98 ± 0.51	ns	3.19 ± 0.26	2.97 ± 0.34	ns
	9 ± 0.5	3.44 ± 0.92	7.05 ± 1.42	**	3.72 ± 0.76	7.77 ± 0.32	***	3.26 ± 0.39	6.30 ± 0.52	***	3.24 ± 0.36	6.25 ± 0.37	***	3.39 ± 0.42	4.65 ± 1.22	ns
	Sig.	ns	**		ns	***		ns	***		ns	***		ns	*	
14	4 ± 0.5	-	-	-	3.65 ± 0.12	2.66 ± 0.54	*	3.27 ± 0.40	3.50 ± 0.62	ns	2.97 ± 0.42	3.04 ± 0.37	ns	3.12 ± 0.40	3.19 ± 0.39	ns
	9 ± 0.5	-	-	-	3.67 ± 0.56	7.84 ± 0.15	***	3.24 ± 0.54	6.59 ± 0.77	***	3.31 ± 0.33	6.65 ± 0.38	***	3.28 ± 0.33	4.90 ± 1.48	ns
	Sig.	-	-	-	ns	***		ns	***		ns	***		ns	ns	
21	4 ± 0.5	-	-	-	3.17 ± 0.16	2.60 ± 0.60	ns	-	-	-	-	-	-	-	-	-
	9 ± 0.5	-	-	-	3.74 ± 0.68	8.35 ± 0.21	***	-	-	-	-	-	-	-	-	-
	Sig.	-	-	-	ns	***		-	-	-	-	-	-	-	-	-
28	4 ± 0.5	-	-	-	2.91 ± 0.22	2.81 ± 0.28	ns	-	-	-	-	-	-	-	-	-
	9 ± 0.5	-	-	-	3.61 ± 0.59	8.76 ± 0.64	***	-	-	-	-	-	-	-	-	-
	Sig.	-	-	-	ns	***		-	-	-	-	-	-	-	-	-

* A.I.—after inoculation; ** N—Normal (pH); Sig.—Significance: ns—not significant; * *p* < 0.05, ** *p* < 0.01 and *** *p* < 0.001.

**Table 2 foods-12-01420-t002:** Counts of *L. monocytogenes* (log CFU/g) in inoculated beef samples according to pH groups (Normal and DFD) in five different packagings and stored at two temperatures (4 ± 0.5 °C and 9 ± 0.5 °C) at day 1, 3, 7, 10, 14, 21 and 28 pm.

Days pm	Temp (°C)	Air			Vacuum			MAP70/20			MAP50/40			MAP30/60		
		N **	DFD	Sig.	N	DFD	Sig.	N	DFD	Sig.	N	DFD	Sig.	N	DFD	Sig.
1	A.I.*	3.10 ± 0.88	3.02 ± 1.12	ns	2.90 ± 0.64	3.13 ± 0.81	ns	2.96 ± 0.94	3.02 ± 0.75	ns	2.91 ± 0.96	3.06 ± 0.82	ns	2..46 ± 1.07	2.97 ± 0.99	ns
3	4 ± 0.5	2.78 ± 1.19	2.91 ± 1.15	ns	2.80 ± 1.21	3.20 ± 0.86	ns	2.64 ± 0.96	2.64 ± 0.89	ns	2.56 ± 0.89	2.60 ± 0.98	ns	2.34 ± 1.29	2.55 ± 1.21	ns
	9 ± 0.5	2.81 ± 0.93	3.39 ± 1.05	ns	2.95 ± 1.04	3.74 ± 1.17	ns	2.61 ± 0.65	3.41 ± 0.92	ns	2.61 ± 0.79	3.17 ± 1.00	ns	2.74 ± 0.66	3.19 ± 0.92	ns
	Sig.	ns	ns		ns	ns		ns	ns		ns	ns		ns	ns	
7	4 ± 0.5	2.51 ± 0.90	3.76 ± 0.90	ns	2.71 ± 0.92	3.86 ± 0.92	ns	2.21 ± 0.93	3.06 ± 0.69	ns	2.08 ± 0.87	2.92 ± 0.81	ns	2.25 ± 0.66	2.64 ± 0.99	ns
	9 ± 0.5	1.83 ± 0.67	7.25 ± 0.68	***	2.68 ± 0.83	7.11 ± 0.61	***	1.74 ± 0.76	6.49 ± 0.83	***	1.79 ± 0.91	6.17 ± 0.91	***	1.99 ± 0.77	5.55 ± 0.62	***
	Sig.	ns	***		ns	**		ns	***		ns	**		ns	**	
10	4 ± 0.5	2.48 ± 0.62	4.47 ± 0.91	*	2.55 ± 1.10	4.55 ± 0.90	*	1.59 ± 0.88	3.89 ± 0.90	*	1.63 ± 1.08	3.21 ± 0.79	ns	1.49 ± 0.85	2.63 ± 0.98	ns
	9 ± 0.5	2.07 ± 0.75	8.25 ± 0.45	***	2.54 ± 0.81	8.24 ± 0.20	***	1.69 ± 1.01	7.91 ± 0.57	***	1.37 ± 0.74	7.79 ± 0.20	***	1.42 ± 0.72	7.01 ± 0.37	***
	Sig.	ns	***		ns	***		ns	***		ns	***		ns	***	
14	4 ± 0.5	-	-	-	2.04 ± 1.04	5.71 ± 0.81	**	1.35 ± 0.79	4.82 ± 1.05	**	1.32 ± 0.72	4.07 ± 1.29	**	1.68 ± 1.15	3.33 ± 1.27	ns
	9 ± 0.5	-	-	-	2.66 ± 0.53	8.33 ± 0.13	***	2.50 ± 0.78	8.55 ± 0.32	***	2.54 ± 0.74	8.53 ± 0.30	***	2.45 ± 0.75	7.91 ± 1.09	***
	Sig.	-	-	-	ns	***		ns	***		ns	***		ns	**	
21	4 ± 0.5	-	-	-	2.08 ± 0.64	6.78 ± 0.69	***	-	-	-	-	-	-	-	-	-
	9 ± 0.5	-	-	-	2.23 ± 0.72	8.29 ± 0.28	***	-	-	-	-	-	-	-	-	-
	Sig.	-	-	-	ns	**		-	-	-	-	-		-	-	-
28	4 ± 0.5	-	-	-	1.71 ± 1.20	7.57 ± 0.41	***	-	-	-	-	-		-	-	-
	9 ± 0.5	-	-	-	1.86 ± 0.69	8.14 ± 0.41	***	-	-	-	-	-		-	-	-
	Sig.	-	-	-	ns	ns		-	-	-	-	-		-	-	-

* A.I.—after inoculation; ** N—Normal (pH); Sig.—Significance: ns—not significant; * *p* < 0.05, ** *p* < 0.01 and *** *p* < 0.001.

## Data Availability

The data presented in this study are available on request from the corresponding author.

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
