# Peer review of "Behaviour of *Escherichia coli* O157:H7 and *Listeria monocytogenes* in Normal and DFD Beef of an Autochthonous Portuguese Breed"

_foods, 2023, doi:10.3390/foods12071420_

Round 1

Reviewer 1 Report

The authors have presented an interesting study and the results are easy to understand. However, the English language must be improved.

Page 1 Abstract line 5: replace "beef meat" with "beef".

line 18.  (ca. 2 or 3 log.. ) must be replaced with (2-3 log..)

rephrase " ...of those both .."

line 23. rephrase last sentence in abstract.

line 28-29. improve the start of the introduction.  (suggest "Beef from cattle of M. breed has high commercial value due to..")

line 32. add "it" in "it is important.."

Page 2 line 1-2. rephrase or delete this sentence.

line 84. Suggest "days of storage" in stead of "days p.m."

line 90. rephrase "bulls with 9 to 11 months of age"

line 95. The beef samples were frozen before use. How was the thawing performed? add a description

page 3 line 100. " using to two"and "per each"?

line 128. Rephrase "count have not been carried out"

Table 1: how many samples were analysed in each group?

line 152 rephrase "all time of storage"

line 159 rephrase "showed the lowest counts showing"

line 167. replace "on opposite" with "however"

line 216. rephrase "growth along time"

line 218 delete "located"

line 221. replace "results" with "differences"

line 307. maybe "cold chain" is better phrase.

line 310-312. Rephrase and explain the results in a better way.

Author Response

The authors are grateful to the referees their attentive and detailed remarks which helped to considerably improve the paper.

We hope the answers below and modifications introduced in the manuscript are clear and concise enough as required by the Reviewers in order to enable the publication of the manuscript in Foods.

All the revisions are highlighted and marked with blue color in the text.

Reviewer 2 Report

Introduction

Line 74-76: In the production system of Maronesa beef is known the occurrence of DFD condition, with a predictable reduction in the product’s shelf-life and meat characteristics become less acceptable to the consumer [31].

It is surprising how the DFD condition is a problem for a Protected Designation of Origin (PDO);

Authors should show specific data or references about the prevalence of DFD condition in the Maronesa breed; since this is the objective of the study, the sentence is very generic;

Line : 223-224 enough to produce such an extensive inhibition of L. monocytogenes as observed in E. coli  H157:H7.

O157:H7

Line 274: atmosphere with high CO2 content (60%) and E. coli O157:H7 counts do not not attained

delete a not

Conclusions

Since the aim of this study was to evaluate the influence of pHu (Normal and High) and of meat packaging (air, vacuum and three MAPs with gas) on the behavior of inoculated E. coli O157:H7 and L. monocytogenes on Maronesa breed cattle and stored stored at 4±0.5ºC and 9±0.5ºC, during 28 days post mortem (pm),

the conclusions are a bit generic; the authors should refer the conclusions also to the meat of the Maronesa breed, since this was the aim of the study

Author Response

The authors are grateful to the referees for their attentive and detailed remarks which helped to considerably improve the paper.

We hope the answers below and modifications introduced in the manuscript are clear and concise enough as required by the Reviewers in order to enable the publication of the manuscript in Foods.

All the revisions are highlighted and marked with blue color in the text.
